Biomass distribution pattern and stoichiometric characteristics in main shrub ecosystems in Central Yunnan, China

Guo Zihao 1
Chen Wei 2
Chen Qianwei 3
Liu Xingyue 1
Hong Sisi 1
Zhu Xiuwen 1
Gong Hede gonghede3@163.com 1
1 School of Geography and Ecotourism, Southwest Forestry University , Kunming , Yunnan , People’s Republic of China
2 College of Landscape Architecture and Horticulture Science, Southwest Forestry University , Kunming , Yunnan , People’s Republic of China
3 Green Development Research Institute, Southwest Forestry University , Kunming , Yunnan , People’s Republic of China
Muukkonen Petteri
Electronic publication date: 2022 Feb 28
Publication date: 2022
Volume: 10
Electronic Location ID: e13005
Received 2021 Oct 11; Accepted 2022 Feb 3
Copyright: ©2022 Guo et al.
Copyright year: 2022
Copyright holder: Guo et al.
License: This is an open access article distributed under the terms of the Creative Commons Attribution License, which permits unrestricted use, distribution, reproduction and adaptation in any medium and for any purpose provided that it is properly attributed. For attribution, the original author(s), title, publication source (PeerJ) and either DOI or URL of the article must be cited.
License URL: https://creativecommons.org/licenses/by/4.0/

Keywords: Shrub plants, Biomass, Carbon density, Stoichiometry, Environmental factors, Distribution pattern

Funding: The National Natural Science Foundation of China 31560189 National Science and Technology Fundamental Work Special Sub-project Investigation of Major Shrub Plant Communities in my country Sub-project 2015FY110300 Investigation of Shrub Plant Community in Yunnan Province 2015FY1103001-4 This work was supported by the National Natural Science Foundation of China “Research on the Impact of Climate Warming on Rhizosphere Soil Carbon Process in Ailao Mountain Evergreen Broad-leaved Forest (31560189),” the National Science and Technology Fundamental Work Special Sub-project “Investigation of Major Shrub Plant Communities in my country (2015FY110300) Sub-project” and “Investigation of Shrub Plant Community in Yunnan Province (2015FY1103001-4)”. The funders had no role in study design, data collection and analysis, decision to publish, or preparation of the manuscript.

==============================
Background

With the exacerbating effects of the global climate change and the more and more attention to the study of plant carbon sink, an increasing number of researches on plant carbon sinks has grown. Although many studies exist on shrub vegetation, soil and litters, most studies focus on the community structure, biomass, surface soil of single plant and shrub layer vegetation, and lack the studies which included the potential relationships between climate change and ecological stoichiometric elements, comprehensive research on main species, even herb and litter layer. In order to provide relevant theoretical basis and data support, it is necessary to take the main terrestrial shrub ecosystem in Central Yunnan as the starting point to analyze and explore its carbon sink distribution characteristics, formation causes, the correlation between climatic factors (temperature and precipitation) and stoichiometric elements, which from community and species levels.

Methods

Plants which originated from 12 main shrub species, litter and soil samples which collected in 69 plots were from 23 plots (Q1–Q23) of 11 cities (countries) in the central Yunnan, China. The biomass and carbon density distribution pattern of each shrub ecosystem and the potential correlations with main climate factors was explored and identified. Some indexes were analyzed such as biomass and carbon density of each part of the shrub ecosystem distribution pattern, correlation, significant changes, formation reasons with the mean value (±standard deviation: SD). Through the redundancy analysis(RDA) of carbon (c), nitrogen (n), phosphorus (P) and main climate factors (precipitation and temperature), the distribution pattern of stoichiometric elements in shrub ecosystem can be judged.

Results

(1) The above-ground biomass (AGB), under-ground biomass (UGB) and root-shoot ratio (R/S) were between 1.13–2.03 t/hm2, 0.62–1.49 t/hm2, and 0.38–0.84, the carbon element was distributed in herb layer under-ground part and rhizomes of the shrub layer mostly. (2) The fitting slope of AGB and UGB of shrub communities and species was in accordance with the allometric distribution growth relationship, the R/S of shrubs was smaller than other vegetation types. Mean annual temperature (MAT) and mean annual precipitation (MAP) are not the main factors which affect the biomass and R/S. (3) The contents of C, N and P elements in leaves were significantly higher than other parts in shrub layer. P in shrub layer above-ground part is much higher than under-ground part. The surface soil layer has the highest C content, and decreased with the depth, so as the impact of vegetation and litter on the content of soil elements. Both of the correlation of MAT with N content of leaf, C/N of stem, the correlation of MAP with C content, C/N of soil is the greatest.

Introduction

With about 2  × 108 hm2 coverage which accounts for nearly 1/5 of the total land area in China, the shrubs which characterized by a wide variety, distribution and high productivity roles an significant part to the terrestrial ecosystems (Hu et al., 2006; Zhang et al., 2011). As one of the provinces which has the most plant species in China, the shrub area reaches 24% to Yunnan Province, and the carbon reserves account for 32.6%.

There are many studies on the expansion of shrub ecosystem abroad in recent years. Frost found that the expansion of shrubs to grasslands is a trend in the Arctic region (Frost & Epstein, 2014; Myers & Hik, 2013); Jørgensen found that shrublands are expanding in the Greenland Islands (Jørgensen, Meilby & Kollmannet, 2013), and Eldridge et al. (2011) thought that through its influence on the structure and function of the shrub ecosystem, the expansion of the shrub does not necessarily represent the degradation of the ecosystem, but may also suggest the development of the ecosystem. Nie et al. (2018) studied the biomass distribution pattern of the herb layer in the alpine shrub ecosystem of the Qinghai-Tibet Plateau found that there is a power function relationship between AGB and UGB, and there is no significant difference between R/S, MAT, and MAP. (Huang, Gao & Huang, 2021) studied the ecological stoichiometric characteristics of C, N, P in 3 evergreen broad-leaved forests in Fujian, and believed that the lack of soil P was an important factor affecting to the nutrient cycle, after analyzing the chemical elements of 15 plant leaves in the Gonghe Basin, Zhang et al. (2018) concluded that the main limiting factors for their growth process was N and P. At the same time, many experts have studied the productivity of shrub plants and their corresponding contribution to the ecosystem from the aspects of energy substitution, community succession, soil and water conservation and nutrient cycle (Piao, Fang & Huang, 2010; Maestre et al., 2009; Rey, Siles & Alcántara, 2009; Pueyo et al., 2013; Chapin, 1983). But most studies only focus on the community structure, biomass and soil microorganisms (Yang et al., 2013), or the single plant of the shrub layer and shrub plants (Niklas, 2005; Sears et al., 2018), even limited to the changes of the overall characteristics of plant nutrient elements in large-scale geographical environments, and lacked horizontal comparison of the comprehensive characteristics of elements among plant ecosystems in specific regional environments.

Therefore, studying the carbon sink and distribution pattern of the shrub plants in this area is beneficial to the region research which belongs to the global climate change in summary, and the comprehensive analysis of the internal ecosystem and ecological stoichiometric element changes of shrub plants are not only useful to indicate distribution characteristics and diversity changes, but also can predict the impact of climate change on the shrub ecosystem and regional human society.

Materials and Methods

Site description

Though avoiding strong interference areas which according to the principle of relatively unified species composition and community habitat structure, and the shrub community types, areas, distributions which recorded in 1:1000000 Vegetation Atlas of China (China Vegetation Map Editorial Committee, 2001), the sampling points were selected. Surveys were conducted from June to September in 2019, and 23 sampling sites which have 3 repetitions for each (23°57′45″–26°31′36″N, 101°1′40″–104°16′5″E) were arranged in 11 counties in Central Yunnan Province (Fig. 1A). The field investigation was made to determine the actual shrub distribution sampling points, and record the information of the shrubs. Latosol is the mainly soil type in Central Yunnan, which is affected by soil forming parent material and vegetation type. 3 typical plots (5 m × 5 m) which spacing about 5 m were arranged which along the diagonal line of each sample point, according to the distribution characteristics (Fig. 1B), and 1m deep which divided into 0–10, 10–20, 20–30, 30–50, 50–70 and 70–100 cm were excavated with 3 soil drills randomly along the diagonal direction in each sample, then selected a plot (1 m × 1 m) to collect and record the biomass information of the soil layer randomly. Along the other diagonal of the sample point, 3 samples (1 m2) were arranged randomly to collect litter near to 25 m2 plots.

Figure 1 The location of the research site and the layout of the sample plot.

Geographical location and experiment study site. Research location (red triangle) (A); layout of the sample plot (B).

There are 12 main shrub types in the study plot: Myrica nana A. Cheval; Quercus pseudosemecarpifolia A. Camus; Myrsine africana Linn; Phyllanthus emblica Linn; Rhododendron adenogynum Diels; Loropetalum chinense (R. Br.) Oliver; Quercus fabri Hance; Rhododendron telmateium Balf. F. et W. W. Smith; Rhododendron racemosum Franch; Rhododendron delavayi Franch; Pyracantha fortuneana (Maxim.) Li; and Zanthoxylum planispinum Sieb.

Experimental design

Referred to previous experimental research designs (Li et al., 2018; Xue et al., 2020), we used a similar setup for sample collection and weighing.

The various shrub and herb plants were recorded (≥150 g) in each 1 m2 plots and the basic information (species names, plants height numbers, crown widths, phenological periods, etc.) were collected. The plants and litters were weighed after the washed and processed samples baked to a constant weight in a box within 70°C, 48 h, and the litter was weighed in each 1 m2 plot (≥30 g)which adjacent to the other diagonal. Soil samples were brought to the laboratory to measuring its bulk density, organic carbon content and element content. The C, N contents were determined by C/N element analyzer, and P contents were determined by molybdenum blue colorimetry.

Data collection and calculations

All data were expressed as mean ± SD.

The biomass was obtained by multiplying the coverage and the dry weight of the sample: community level biomass = coverage × dry weight, the average (±SD) of the original data was calculated by Microsoft Excel 2105 software. The AGB, UGB, R/S, and the slope and intercept of the data were obtained, then the relationship between the logMA and logMB were established on this basis after the logarithmic transformation of communities, species and main shrubs, and the slope of the regression line to determine whether the linear relationship was constant or allometric was assessed (Nie et al., 2016; Tao & Zhang, 2014).The soil and plant elements their internal distribution relationships was analyzed by Pearson Correlation though the IBM-SPSS Statistics 23 software, the significance of the content and ratio of soil elements in different soil layers was tested by Least Significant Difference (LSD), and the mass fraction of stoichiometric elements was measured in the meantime (Su et al., 2017).

We used the thin plate smoothing spline function algorithm to interpolate the meteorological data to extract and calculate the mean annual temperature (MAT) and mean annual precipitation (MAP) data of the sample plot which was download from the WorldClim website (https://www.worldclim.org/) (Nie et al., 2018), the interaction between the elements of the shrub ecosystem to the temperature and precipitation factors was analyzed by CANOCO 5.0.

Results

Distribution pattern and fitting situation of carbon sink of main shrubs

When the shrub layer grows, the biomass is mostly concentrated in roots and stems, with leaf being the least significant of them, to the herb layer, the biomass is mostly in the under-ground part, and to the litter layer, it is significantly lower than the above which mentioned (Table 1), which means that the main shrubs distribute C throughout the deeper soil layers and in rhizomes during their life cycle in central Yunnan.

The variation range of AGB and UGB at the community level exceeds the species level, but the R/S is opposite, which means the biomass of the main shrubs is mostly concentrated in deeper soil layers, excluding the species characteristics, most shrub plants choose to concentrate the nutrients which is used for growth in rhizomes and other under-ground structures, in order to prevent the loss of nutrient elements which caused by sunlight exposure or insufficient rain. This finding similars to the previous research results, and can help us to identify the spatial distribution pattern of main shrubs in Central Yunnan (Jackson et al., 1996; Hui & Jackson, 2005; Mokany, Raison & Prokushkin, 2006).

Table 1 Biomass of main shrub communities in Central Yunnan t/hm2.

Sample plot	Shrub layer	Herb layer	Litter layer	Biological layer	
	Root	Stem	Leaf	Under-ground	Above-ground			
Q1	0.70 ± 10.38	0.65 ± 7.97	0.50 ± 6.91	0.60 ± 5.60	0.53 ± 4.66	0.01 ± 0.55	2.99 ± 6.01	
Q2	0.62 ± 11.81	0.64 ± 5.22	0.49 ± 3.01	/	/	0.01 ± 0.27	1.76 ± 3.39	
Q3	0.68 ± 11.90	0.65 ± 3.70	0.65 ± 16.57	0.68 ± 1.24	0.64 ± 10.40	0.01 ± 0.10	3.30 ± 7.32	
Q4	0.61 ± 6.78	0.61 ± 8.03	0.52 ± 5.99	0.62 ± 0.00	0.57 ± 0.00	0.01 ± 0.15	2.94 ± 3.49	
Q5	0.63 ± 8.35	0.63 ± 8.35	0.56 ± 3.93	0.14 ± 0.00	0.81 ± 0.00	0.02 ± 1.03	2.80 ± 3.61	
Q6	0.66 ± 7.52	0.62 ± 3.95	0.51 ± 2.29	0.14 ± 0.00	0.12 ± 0.00	0.01 ± 0.18	2.06 ± 2.32	
Q7	0.75 ± 4.51	0.69 ± 5.72	0.45 ± 3.18	0.62 ± 11.30	0.49 ± 8.67	0.01 ± 0.08	3.00 ± 5.58	
Q8	0.62 ± 2.94	0.58 ± 9.38	0.52 ± 4.54	0.55 ± 13.77	0.44 ± 10.54	0.02 ± 0.67	2.72 ± 6.97	
Q9	0.56 ± 4.21	0.56 ± 2.84	0.54 ± 2.81	0.56 ± 0.00	0.58 ± 0.00	0.01 ± 0.35	2.81 ± 1.70	
Q10	0.57 ± 5.12	0.57 ± 28.29	0.54 ± 2.69	0.51 ± 3.95	0.55 ± 16.50	0.01 ± 0.10	2.74 ± 9.44	
Q11	0.76 ± 6.20	0.78 ± 3.24	0.69 ± 3.98	0.51 ± 1.97	0.39 ± 11.71	0.01 ± 0.04	3.14 ± 4.52	
Q12	0.62 ± 2.79	0.64 ± 4.72	0.56 ± 8.73	0.59 ± 6.45	0.52 ± 1.65	0.01 ± 0.10	2.94 ± 4.07	
Q13	0.78 ± 7.31	0.77 ± 4.56	0.54 ± 2.91	0.61 ± 9.81	0.42 ± 5.95	0.01 ± 0.12	3.13 ± 5.11	
Q14	0.71 ± 3.23	0.76 ± 17.28	0.52 ± 3.47	0.59 ± 5.68	0.40 ± 9.40	0.01 ± 0.05	2.970 ± 6.52	
Q15	0.60 ± 4.06	0.68 ± 8.26	0.52 ± 8.12	0.42 ± 1.10	0.21 ± 3.21	0.02 ± 0.21	2.44 ± 4.16	
Q16	0.61 ± 5.87	0.69 ± 6.39	0.47 ± 1.97	0.38 ± 15.32	0.39 ± 3.58	0.02 ± 0.26	2.55 ± 5.57	
Q17	0.70 ± 5.45	0.80 ± 6.13	0.66 ± 0.97	0.54 ± 9.10	0.49 ± 6.83	0.02 ± 0.33	3.22 ± 4.80	
Q18	0.62 ± 0.77	0.65 ± 1.31	0.57 ± 5.54	0.42 ± 11.38	0.33 ± 7.65	0.03 ± 0.23	2.63 ± 4.48	
Q19	0.55 ± 5.49	0.65 ± 2.26	0.34 ± 2.69	0.62 ± 7.32	0.43 ± 3.54	0.04 ± 0.13	2.62 ± 3.57	
Q20	0.71 ± 6.21	0.75 ± 5.50	0.78 ± 4.65	0.50 ± 6.07	0.51 ± 6.45	0.03 ± 0.12	3.28 ± 4.83	
Q21	0.65 ± 8.66	0.75 ± 6.07	0.56 ± 6.64	0.37 ± 6.21	0.30 ± 6.94	0.03 ± 0.06	2.67 ± 5.76	
Q22	0.75 ± 6.63	0.83 ± 2.26	0.54 ± 8.17	0.63 ± 6.40	0.43 ± 12.33	0.03 ± 0.05	3.20 ± 5.97	
Q23	0.79 ± 2.73	0.66 ± 3.59	0.60 ± 13.82	0.70 ± 1.79	0.62 ± 6.75	0.03 ± 0.07	3.40 ± 4.79	
Notes.

A slash (/) indicates no samples were collected.

Table 2 Community and species for biological layer biomass and R/S of main shrubs in Central Yunnant/hm2.

Level		Biomass	Species name	Sample size	R2	Slope	95% confidence interval	Intercept	
		Min	Max	Mean							
Community(C)	AGB	1.13	2.03	1.67		69	0.367	0.422	0.170–0.674	0.195	
UGB	0.62	1.49	1.15		
R/S	0.38	0.84	0.69		
Species(S)	AGB	0.98	1.67	1.28		64	0.194	0.586	0.083–1.088	0.199	
UGB	0.57	0.82	0.69		
R/S	1.37	2.34	1.86		
S1	Myrica nana A. Cheval	23	0.162	0.456	−0.014–0.927	0.170	
S2	Quercus pseudosemecarpifolia A. Camus	3	0.150	0.091	−2.667–2.849	0.225	
S3	Myrsine africana Linn	5	0.354	−0.327	−1.139–0.484	−0.007	
S4	Phyllanthus emblica Linn	3	0.036	−0.095	−6.383–6.193	0.155	
S5	Rhododendron adenogynum Diels	3	0.027	−0.115	−8.862–8.631	0.048	
S6	Loropetalum chinense (R. Br.) Oliver	3	0.942	0.875	−1.876–3.626	0.154	
S7	Quercus fabri Hance	3	0.051	0.500	−27.010–28.010	0.195	
S8	Rhododendron telmateium Balf. F. et W. W. Smith	3	0.001	0.019	−6.329–6.368	0.166	
S9	Rhododendron racemosum Franch	6	0.049	−0.185	−1.313–0.943	0.036	
S10	Rhododendron delavayi Franch	3	0.750	−3.000	−25.008–19.008	−0.330	
S11	Pyracantha fortuneana (Maxim.) Li	3	0.429	0.410	0.867–0.545	0.149	
S12	Zanthoxylum planispinum Sieb	6	0.794	0.703	0.207–1.199	0.190	

There is a positive correlation between AGB and UGB (Table 2), which distributed at the community level (R2 = 0.367, Fig. 2A) has better fitting effect than the species level (R2 = 0.194, Fig. 2B). The slope of the fitting regression line of community level is 0.422 (95% confidence interval is 0.170–0.674,), and 0.586 for the slope of horizontal fitting regression curve of shrub species, which means the main shrub groups in central Yunnan conform to the allometric growth theory. Among the distribution of AGB and UGB (Fig. 2), the fitting effect of S6 (R2 = 0.942, Fig. 2H), S10 (R2 = 0.750, Fig. 2L), S11 (R2 = 0.429, Fig. 2M) and S12 (R2 = 0.794, Fig. 2N) are better than S1 (R2 = 0.162, Fig. 2C), S2 (R2 = 0.150, Fig. 2D), S8 (R2 = 0.001, Fig. 2J) and S9 (R2 = 0.049, Fig. 2K). The slope of S3 (Fig. 2E, 95% confidence interval: −1.139–0.484, p <0.01), S4 (Fig. 2F, 95% confidence interval: −6.383–6.193, p < 0.01), S5 (Fig. 2G, 95% confidence interval: −8.862–8.631, p < 0.01), S9 (Fig. 2K, 95% confidence interval: −1.313–0.943, p < 0.01) and S10 (Fig. 2L, 95% confidence interval of −25.008–19.008, p < 0.01) are negative, which means that the distribution relationship of AGB and UGB is negatively correlated with allometric distribution mode at the species level after calculation and analysis in SPSS.

Figure 2 Distribution relationship between AGB (LogMA) and UGB (LogMB) of main terrestrial shrubs in central Yunnan.

The distribution relationship between AGB (LogMA) and UGB (LogMB) of main terrestrial shrubs in central Yunnan, the order is (A) Community; (B) Species; (C) Myrica nana A. Cheval; (D) Quercus pseudosemecarpifolia A. Camus; (E) Myrsine africana Linn; (F) Phyllanthus emblica Linn; (G) Rhododendron adenogynum Diels; (H) [Loropetalum chinense (R. Br.) Oliver]; (I) Quercus fabri Hance; (J) Rhododendron telmateium Balf. F. et W. W. Smith; (K) Rhododendron racemosum Franch; (L) Rhododendron delavayi Franch; (M) [Pyracantha fortuneana (Maxim.) Li] and (N) Zanthoxylum lanispinum Sieb. Et Zucc.

MAT (Figs. 3A, 3B), MAP (Figs. 3C, 3D), R/S (Figs. 3E, 3F) were not significantly different with AGB and UGB (p < 0.05), which indicates that both MAT and MAP were not the main factor to affect the biomass and R/S of major terrestrial shrubs in Central Yunnan.

Figure 3 Relationship between AGB, UGB, R/S and climate factors of main shrubs in Central Yunnan.

The relationship between AGB, UGB, R/S and climate factors of main shrubs in Central Yunnan. Correlation among various factors, AGB, Above-ground biomass (t/hm2); UGB, Under-ground biomass (t/hm2); MAT, Mean annual temperature (°C); MAP, mean annual precipitation (mm); R/S, Root shoot ratio.

Distribution pattern of main elements in each layer of shrub plants

There are obvious differences to the distribution characteristics of the nutrient elements of the plants which is required for growing of the main shrub plants in central Yunnan (Table 3), which indicates that the parts which performing different functions, adhere to the different distribution patterns. The influence of N, P for the composition of plants is less significant than C. The nutrient elements of the plants are mostly concentrated in the leaves in the shrub layer, and in the above-ground parts in the herb layer.

Table 3 Statistical characteristics of C, N, P element contents stoichiometry of main shrubs in Central Yunnan.

Investigation level	Position	C element content/%	N element content/%	P element content/%	C/N	C/P	N/P	
Shrub layer	Root	45.56	0.40	0.53	125.00	146.20	1.20	
Stem	45.87	0.43	0.48	116.00	131.50	1.20	
Leaf	47.52	1.47	1.17	34.40	54.10	1.60	
Herb layer	Above-ground	40.98	1.08	1.51	43.80	40.43	0.88	
Under-ground	36.25	0.61	1.00	65.50	47.99	0.72	
Litter layer	41.34	1.02	0.81	43.87	68.18	1.48	

Except the significant positive correlation between stem N and leaf N (p < 0.05, Table 4), there was an extremely significant positive correlation between the same elements which located in stems and leaves in shrub layer (p < 0.01) though the Pearson correlation analysis. On the contrary, most different elements has the negative correlation relationships (p < 0.01), which means the influence level and distribution of the same elements are the same. There is a significant negative correlation between root C and leaf P, root N and root P, leaf N and leaf P (p < 0.01), and significant positive correlation between root P and leaf N (p < 0.05), which means the distribution and influence level of nutrient elements between roots and leaves are different. In addition, there was an extremely significant negative correlation between stem C and leaf P, stem P and leaf C, leaf C and leaf P (p < 0.01); an extremely significant positive correlation between stem N and leaf N (p < 0.05); a significant negative correlation between stem C and leaf N (p < 0.05), which means the distribution and influence level of nutrient elements between stems and leaves are different.

Table 4 Correlation of C, N, P contents in different parts of the main shrubs in central Yunnan.

Layer	Part	Element	Root	Stem	Leaf	Soil	
			C	N	P	C	N	P	C	N	P	C	N	P	
Shrub layer	Root	C	1	−0.604∗∗	−0.593∗∗	0.905∗∗	−0.097	−0.577∗∗	0.894∗∗	−0.608∗∗	−0.545∗∗	0.057	−0.001	−0.004	
N		1	0.266	−0.459*	0.757∗∗	0.266	−0.592∗∗	0.783∗∗	0.304	0.478	0.001	0.040	
P			1	−0.569∗∗	−0.067	0.842∗∗	−0.532∗∗	0.512*	0.615∗∗	0.182	0.007	−0.137	
Stem	C				1	0.040	−0.650∗∗	0.816∗∗	−0.506*	−0.605∗∗	0.103	0.002	0.003	
N					1	−0.504	−0.157	0.423*	−0.060	1.180	0.017	0.105	
P						1	−0.597∗∗	0.382	0.787∗∗	−0.002	−0.001	0.142*	
Leaf	C							1	0.553∗∗	−0.668∗∗	0.042	−0.001	0.026	
N								1	0.552∗∗	0.205	0.050	0.520	
P									1	−0.732	0.075	0.512∗∗	
Herb layer			Above-ground	Under-ground			
Above-ground	C	1	−0.203	−0.468*	0.216	0.017	0.199				−0.019	0.031	0.075	
N		1	0.694∗∗	0.197	0.461*	0.420				−0.494	0.446	1.127	
P			1	0.078	0.254	1.000∗∗				−0.620	0.108	0.618∗∗	
Under-ground	C				1	0.142	−0.090				0.283	0.044∗∗	0.059*	
N					1	0.578∗∗				3.861	0.651∗∗	0.917*	
P						1				0.771	0.129*	0.431∗∗	
Litter layer	C		−0.129	0.044∗∗	0.070∗∗	
		N										−2.665	0.682∗∗	1.089∗∗	
		P										−2.058*	0.190∗∗	0.435∗∗	
Notes.

* Significant correlation at the level of 0.05.

** Extremely significant correlation at the level of 0.01.

The analysis is emphasized because the C, N, P of surface soil layer is the most activity to plants. The P in surface soil has significant positive correlation with the stems P and leaves P (p < 0.05). There is an extremely significant positive correlation between surface soil P of the herb layer and in AGB (p < 0.01), and the under-ground N, P has significant positive correlation with surface soil C, N and P respectively (p < 0.05).

The C content is mainly concentrated in the 0–30 cm soil layer which means the highest in the surface soil layer, and gradually decreases with soil depth (Fig. 4). With the soil depth increases, soil C may be lost due to the the influence of runoff or sunshine factors. With the increase of soil depth, N content gradually decreases, while P tends to be stabled, indicating that the fixation of P element has no undulate with the depth.

Figure 4 4 C, N, P content of main shrub soil in central Yunnan.

C, N, P content of main shrub soil in central Yunnan.The C content is mainly concentrated in the 0–30 cm soil layer which too large difference in standard deviation indicates instability, and N content gradually decreases, P tends to be stabled with the soil depth increases.

Redundant analysis of main elements content and ratio

MAT has the greatest correlation with root P, and the MAP has the most significant correlation with litter P, which could reflect the relationship between the stoichiometric characteristics of shrub plants and non- biological factors (Fig. 5A). According to RDAII, the angle of precipitation and P, N and C is less than 90, revealing a positive correlation, and the projection is longer which means the positive correlation has large impact.The angle between the precipitation and root S, leave C, litter layer C is much more than 90° in shrub layer which means negative correlation, and the longer projection shows this factor has significant negative correlation. According to RDA, the angle between MAT, herb C, shrub leaf N, and shrub P are less than 90° which means that temperature has a positive correlation with these. The temperature factor of shrub, litter, soil layer related to C content; N content in root, stem, herb layer, litter layer and shrub layer, P in herb layer, litter layer and soil layer was negatively correlated, and P content of herb layer, litter layer, soil layer is negatively correlated, and with the longer the projection, the temperature factor for negative correlation has more influence. The projection of the temperature is longer which means the positive correlation has significant influence.

Figure 5 Redundant analysis of carbon, nitrogen and phosphorus elements with main environmental factors of main shrubs in central Yunnan.

The redundant analysis of carbon, nitrogen and phosphorus elements with main environmental factors of main shrubs in central Yunnan. Element content (A), mean annual temperrature: E = 11.2%, p = 0.008, Mean annual precipitation: E = 13.9%, p = 0.004; element ratio (B), mean annual temperrature: E = 6.4%, p = 0.196, mean annual precipitation: E = 7.6%, p = 0.136.

The MAT has the largest correlation with the C/N of the soil layer, and the MAP has the largest correlation with the C/P of the under-ground herb layer (Fig. 5B). Along the RDA, the angle between the MAP and the C/N of the soil layer is less than 90° which means positive correlation. Among the main shrubs in central Yunnan, MAT has greater impact than MAP, suggested that temperature is the main factor, and the impact of precipitation on them is limited in the shrub vegetation ecosystem. The angle between the ratio of other elements and the precipitation factor is more than 90° which revealed a negative correlation. Along the RDA, MAT is positively correlated with the C/N of the shrub layer roots, the stem C/N, the herb layer C/P, C/N, N/P, and the litter layer N/P. The angles between the MAT and the N/P, C/P in the roots of the shrubs, the N/P, C/P of the stems, the C/P, N/P, C/N of the leaves N, the C/N, C/P, N/P of the soil layer are all more than 90° , and there is a negative correlation between the MAT and projection, which indicated the negative correlation has a significant influence.

Discussion and Conclusions

Distribution pattern and allocation method of main shrub biomass

By exploring the distribution of biomass at 2 different levels (community and species), after establishing logarithmic transformation, both AGB and UGB have allometric relationship. which is different from Nie’s conclusions (Nie et al., 2016; Wang et al., 2010; Yang et al., 2009). These differences may be caused by the temperature, dry humidity, and differences between AGB and UGB in central Yunnan, and the correlation between the AGB and UGB of the main terrestrial shrub vegetation species is not significantly correlated with the difference in the fitting situation (Tao & Zhang, 2014), which means AGB and UGB will not affect the distribution of the main shrubs in central Yunnan. The R/S of main shrub communities is lower than that of the terrestrial shrub communities of high mountains area in central Yunnan (Mokany, Raison & Prokushkin, 2006), which may be due to the influence which affected by high altitude and dry humidity of the survey area, and higher than the forest and shrub vegetation in Northeast China (Ma et al., 2014), lower than the alpine shrub community, lower than the grassland and forest shrubs in China (Yang et al., 2009). This may be due to different species which affected by different plants under similar latitude and climate conditions in this region, which resulted in an increase in the possibility of having the same distribution pattern to above and under the ground. The same allometric growth distribution mode to the biological and community level indicates that the R/S of grass and shrub is lower than other regions, which extrapolated by distributing biomass to the ground, the maximum growth rate can be obtained during the formation of high-altitude shrubs in the plateau.

The R/S of the vegetation in central Yunnan is small, indicating that the shrub root system is more developed than the above-ground part in central Yunnan, which means the vegetation transferred and distributed more biomass through carbon sinks, compared with other regions. In addition, the sufficient sunshine and insufficient precipitation makes the biomass of under-ground more than above-ground, which create the vegetation invests more biomass to under-ground to obtain more water and nutrients to achieve the maximum growth rate, and the R/S has significant impact on the distribution relationship of vegetation than the significant difference in the fitting slope has an impact on the distribution relationship of vegetation in the data conclusions obtained in this experiment

Relationship between C, N, P stoichiometry and abiotic driving factors in main shrubs

The C, N and P of main shrubs are within the normal range in central Yunnan (Marschner, 1995), however the difference from previous relevant research conclusions is the contents of C, N and P in shrub leaves are higher than those in other levels. The herb layer P is mostly concentrated in above-ground part, which is higher than it in shrub leaves, this may be related to the disturbance of the investigated regional environment by certain human factors and the differential distribution of shrub ecosystems to adapt to the regional environment, which is the same as the temperature, precipitation and light factors.The reason for the N of main shrub lower than the average N, P of Chinese shrub plants, may be the long-term biological evolution and environmental adaptation which leads the formation of different forms of N and P in soil, and the different N absorption mechanisms and plant strategies in central Yunnan (He et al., 2008; Zheng et al., 2007; Zhao, Huang & Ma, 2013). Leaf N/P is an important index of ecological significance, which reflects the structural and functional characteristics of vegetation. Different from previous studies on woody plants (Wu & Ma, 2009), the C content is higher, compared with plant N and P. The difference between N and P is not significant, but C content is opposite which means the unique plateau low latitude climate and sufficient light make large amounts of pivotal organic nutrients for plants.

The N content in soil surface layer is higher than that in other layers. With increasing of soil depth, P shows no significant change which means it is mainly determined by both soil type and parent material (Li, Shi & Chen, 2011). The vegetation and litter contribute the most to soil C and N content, but gradually decreases with soil depth, which consistent with the results of Su’s research (Su et al., 2017) and expressing that the main shrub plants have the same influence characteristics to soil chemical elements in central Yunnan. There is a negative correlation between soil N content and soil depth. However, soil P does not change with soil depth, which is different from Pan et al. (2014); this may be related to the unique rock, soil structure and natural conditions of sunlight and precipitation in the region, and P is concentrated in the deep layer after surface soil loss. The P has an extremely significant positive correlation with leaf P and stem P, which may be due to the absorption of phosphorus by plants from the soil and then transferred to different organs (Zhou et al., 2005). There is a significant correlation between plant N content and soil N, which is because plant nitrogen content is more likely to be a species characteristic than a soil nutrient (He et al., 2010).

Temperature and precipitation are two major environmental factors which affect the growth of shrub plants. Leaf N, leaf P are significant positive correlated with the MAT, while soil N is significant negative correlated with MAT, which is similar to the research results of Jiang and Ma’s (Jiang et al., 2017; Ma et al., 2015), which means the increase of temperature is beneficial for plants to absorb N and P in soil. In addition, the effect of precipitation on the element content of shrub vegetation ecosystem exceeded the impact of temperature factor, and the C/N, C/P, N/P of litter in this area were significantly negative correlated with temperature factor, which is similar to the results of Ge & Xie (2017).

However, as proposed in the study on ecostoichiometric characteristics of C, N, P in Fujian evergreen broad-leaved forest, the C/N, C/P, and N/P of the shrub layer leaves are significantly negative correlated with temperature factors, which contradicts the conclusion that the element ratio of shrub vegetation decreases with the increase in latitude, which may be related to factors such as plant composition and complex life forms. Excepting the significant negative correlation between root C and shrub root leaf, root C and litter, C has a positive significant correlation with the elements content in the shrub vegetation ecosystem, not with the element ratio. Previous studies rarely involved the correlation between herb, litter layer with temperature and precipitation factors, C/N, C/P, N/P of herb layer and litter layer are positive correlated with the MAT in this study, which means the characteristics of plant N and P on the regional scale can reflect the characteristics of plants and their long-term response to environmental conditions to a certain extent.

Supplemental Information

Supplemental Information 1 Biomass of main shrub communities in Central Yunnan

Carbon content of main shrub biosphere in Central Yunnan, Carbon density and distribution pattern of components in main shrub ecosystems in Central Yunnan (mean ± standard deviation), Community level biomass and fitting, and Species biomass and fitting SPSS analysis

Click here for additional data file.

Supplemental Information 2 Raw biomass data of shrub layers

Click here for additional data file.

Supplemental Information 3 Raw data for Figure 4

Click here for additional data file.

Supplemental Information 4 Element content and ratio redundancy analysis

Click here for additional data file.

Additional Information and Declarations

Competing Interests

Author Contributions

Data Availability

The authors declare there are no competing interests.

Zihao Guo conceived and designed the experiments, performed the experiments, analyzed the data, prepared figures and/or tables, and approved the final draft.

Wei Chen conceived and designed the experiments, analyzed the data, prepared figures and/or tables, and approved the final draft.

Qianwei Chen analyzed the data, authored or reviewed drafts of the paper, and approved the final draft.

Xingyue Liu, Sisi Hong and Xiuwen Zhu performed the experiments, analyzed the data, prepared figures and/or tables, and approved the final draft.

Hede Gong conceived and designed the experiments, authored or reviewed drafts of the paper, and approved the final draft.

The following information was supplied regarding data availability:

The raw data is available in the Supplemental File.

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
