# Peer review of "Biomass distribution pattern and stoichiometric characteristics in main shrub ecosystems in Central Yunnan, China"

_PeerJ, doi:10.7717/peerj.13005_

## Round 0.1 · original submission · Major Revisions

This manuscript fits fine under the scope of this journal.

Unfortunately, the current stage and form of the manuscript are not high enough. My decision is that this manuscript needs major revisions and a new review round. Overall, there are many things that are not following the styles or traditions of scientific English publishing. Consider all comments and suggestions given by reviewers.

First, this MS needs a proper language revision. In addition, your text's fluency should be improved.

Improve your Results section according to the proposals given by reviewers. And rewrite the Materials and Methods according to the proposals given by reviewers.

Consider the amount of tables and figures. The number of especially figures is too high.

Detailed comments from the editor:
- Why you are using a symbol ~ between numbers (e.g. 70~100 cm; 71 23°57'45"~ 26°31'36"N) instead of typical symbol - (en dash)? This is not a standard style.
- Check the style of scientific species names --> not everything in italics (author names).
- Table 1. Are numbers in columns in right order? For example 0.70±10.38 Should these be opposite? How can a base number for biomass be near zero while the range is more than 10?
- What unit is t/hm2. This is not a standard style.
- Table 2. Why do you have a range for mean values? And what kind of range values these are? Unclear for the reader.
Table 4. What are these ± values in the table? How you have calculated those and what those are meaning?
- Figure 2. Don't use numerical scales. Use scale bars. When printing your maps in different size these numerical scales are not true anymore.

Reviewer 1 ·

Basic reporting

no comment

Experimental design

no comment

Validity of the findings

Precipitation and temperature serve as important data for the paper,But not found it in the paper, whether it can explain the situation ?In addition, only the average annual rainfall and temperature to analyze the sample data once, the analysis results are not convincing?

Additional comments

1,The sampling point Q1-Q23, is not clearly annotated in Figure 2
2.In Table 2, the standard deviation of the data is too large, such as "0.70±10.38" .Can you explain these standard deviations?

Annotated reviews are not available for download in order to protect the identity of reviewers who chose to remain anonymous.

Reviewer 2 ·

Basic reporting

I commend the authors for their extensive data set, compiled over large number detailed fieldwork. The results are interesting and useful to future readers. However, the manuscript need further modification, especially for the section of the results.
1. The most important issue.
The results should be rewritten. The most important results (such as biomass pattern, stoichiometric characteristics, and their relationship with MAP and MAT).
2. The next most important item
The language should be edited in the entire manuscript. Many sentences are difficult to understand. Some examples where the language could be improved include Lines 8-10, 13, 16-18, 38-39, 65-67, …….
I strongly suggest you have a colleague who is proficient in English and familiar with the subject matter review your entire manuscript, or contact a professional editing service.
3. The third important points.
The part of MATERIALS & METHODS should be rewritten. The section of Site description and Experimental design are too repetitious.
4. The fourth important points
Previous studies about the shrub land in the study sites should be added in the introduction.
5. The fifth important points
Too many figures and tables. Delete or combine some figures and tables. Just show the key resuls.

Experimental design

no comment

Validity of the findings

no comment

Additional comments

Other point
1. Check the Latin scientific names of animals, plants and microorganisms in the text (subject to the latest authoritative reference book), legal unit of measurement symbols, etc;
2. Detailed information needs to be supplemented. What is the basis, representativeness and the main shrub type?
3. How to collect root, stem and leaf samples of shrub layer?
4. The text is rough, many places are not smooth, the abstract is not concise, and the writing and terminology are not standardized. The decimal places in the chart are not uniform and need to be corrected;
6. The length of the article is too long. Please reduce the length appropriately. The total number of characters shall not exceed 12000 words (including spaces), and the total page number of word documents shall not exceed 10 pages (including charts).

Some other suggestions and comments are edited in the manuscript.

Annotated reviews are not available for download in order to protect the identity of reviewers who chose to remain anonymous.

Reviewer 3 ·

Basic reporting

Shrub ecosystem plays vital roles in terrestrial ecosystems. Past studies mainly focused on community structure and biomass, or single plant of the shrub layer and shrub plants, but still lacked horizontal comparison of elements and the relationship between climatic factors. Guo et al. selected 12 main shrub types in the study area, and obtained soil and vegetation data. I have found the present manuscript is interesting and has great application. The study is impressive in scale and appears to provide some interesting insight to the distribution relationship in the terrestrial shrub ecosystem and the relationship between climatic factors. Overall, the text of the manuscript is well-organized but there are some areas that require revision for clarity. I hope my comments are helpful to the authors in revising their manuscript. In addition, The English language should be improved to ensure that an international audience can clearly understand your text. I suggest you have a colleague who is proficient in English and familiar with the subject matter review your manuscript, or contact a professional editing service.
Lines 14-15 This sentence need to rewrite.
Lines 28-31 Do you have some more summary conclusions?
Lines 60-67 For this section, you can put forward several assumptions or questions.
Lines 70-31 “Surveys were conducted from June to September, 2019”, why not in a same time?
Line 132 There are so many results, and you need to divide them into different sections.
Line 234 Like results section, you need to divide them into different sections.
In addition, there are so many tables and figures, you need put some tables or figures in the appendixes.

Experimental design

This is an original primary research within aims and scope of the journal. You can put forward several assumptions or questions in Introduction section.

Validity of the findings

All underlying data have been provided, and they are robust, statistically sound & controlled.

Annotated reviews are not available for download in order to protect the identity of reviewers who chose to remain anonymous.

---

## Round 0.2 · Minor Revisions

Dear authors!

One reviewer and I have checked your revised manuscript and we see that you have improved your manuscript a lot. A few minor style issues popped up. The manuscript is now much clearer and the English is much better too.

1) You are missing a space in a few places just before parentheses. Check them all.

2) Indicator for p-value is not with a capital P but lower case "p".

3) The sub-heading "Redundant analysis of main elements in each layer of shrub plants with temperature and precipitation factors" is too long.

4) Check also literature citations in the text. Sometimes you use the "&" mark and sometimes not. In addition, you are missing some spaces also here. Check every citation in the text and use a consistent style.

Reviewer 3 ·

Basic reporting

The manuscript has been modification.

Experimental design

No comment.

Validity of the findings

No comment.

Additional comments

No comment.

---

## Round 0.3 · accepted · Accept

You have made those minor changes requested. Thank you for that.